# Linear Transformers are Versatile In-Context Learners

**Max Vladymyrov**
Google Research
mxv@google.com

**Johannes von Oswald**
Google, Paradigms of Intelligence Team
jvoswald@google.com

**Mark Sandler**
Google Research
sandler@google.com

**Rong Ge**
Duke University
rongge@cs.duke.edu

## Abstract

Recent research has demonstrated that transformers, particularly linear attention models, implicitly execute gradient-descent-like algorithms on data provided in-context during their forward inference step. However, their capability in handling more complex problems remains unexplored. In this paper, we prove that each layer of a linear transformer maintains a weight vector for an implicit linear regression problem and can be interpreted as performing a variant of preconditioned gradient descent. We also investigate the use of linear transformers in a challenging scenario where the training data is corrupted with different levels of noise. Remarkably, we demonstrate that for this problem linear transformers discover an intricate and highly effective optimization algorithm, surpassing or matching in performance many reasonable baselines. We analyze this algorithm and show that it is a novel approach incorporating momentum and adaptive rescaling based on noise levels. Our findings show that even linear transformers possess the surprising ability to discover sophisticated optimization strategies.

## 1 Introduction

The transformer architecture (Vaswani et al., 2017) has revolutionized the field of machine learning, driving breakthroughs across various domains and serving as a foundation for powerful models (Anil et al., 2023; Achiam et al., 2023; Team et al., 2023; Jiang et al., 2023). However, despite their widespread success, the mechanisms that drive their performance remain an active area of research. A key component of their success is attributed to in-context learning (ICL, Brown et al., 2020) – an emergent ability of transformers to make predictions based on information provided within the input sequence itself, without explicit parameter updates.

Recently, several papers (Garg et al., 2022; Akyürek et al., 2022; von Oswald et al., 2023a) have suggested that ICL might be partially explained by an implicit meta-optimization of the transformers that happens on input context (aka mesa-optimization Hubinger et al., 2019). They have shown that transformers with linear self-attention layers (aka linear transformers) trained on linear regression tasks can internally implement gradient-based optimization.

Specifically, von Oswald et al. (2023a) demonstrated that linear transformers can execute iterations of an algorithm similar to the gradient descent algorithm (which they call $GD^{++}$), with each attention layer representing one step of the algorithm. Later, Ahn et al. (2023); Zhang et al. (2023) further characterized this behavior, showing that the learned solution is a form of preconditioned GD, and this solution is optimal for one-layer linear transformers.

38th Conference on Neural Information Processing Systems (NeurIPS 2024).

In this paper, we continue to study linear transformers trained on linear regression problems. We prove that each layer of every linear transformer maintains a weight vector for an underlying linear regression problem. Under some restrictions, the algorithm it runs can be interpreted as a complex variant of preconditioned gradient descent with momentum-like behaviors.

While maintaining a linear regression model (regardless of the data) might seem restrictive, we show that linear transformers can discover powerful optimization algorithms. As a first example, we prove that in case of GD$^{++}$, the preconditioner results in a second order optimization algorithm.

Furthermore, we demonstrate that linear transformers can be trained to uncover even more powerful and intricate algorithms. We modified the problem formulation to consider mixed linear regression with varying noise levels[1] (inspired by Bai et al., 2023). This is a harder and non-trivial problem with no obvious closed-form solution, since it needs to account for various levels of noise in the input.

Our experiments with two different noise variance distributions (uniform and categorical) demonstrate the remarkable flexibility of linear transformers. Training a linear transformer in these settings leads to an algorithm that outperforms GD$^{++}$ as well as various baselines derived from the exact closed-form solution of the ridge regression. We discover that this result holds even when training a linear transformer with diagonal weight matrices.

Through a detailed analysis, we reveal key distinctions from GD$^{++}$, including momentum-like term and adaptive rescaling based on the noise levels.

Our findings contribute to the growing body of research where novel, high-performing algorithms have been directly discovered through the reverse-engineering of transformer weights. This work expands our understanding of the implicit learning capabilities of attention-based models and highlights the remarkable versatility of even simple linear transformers as in-context learners. We demonstrate that transformers have the potential to discover effective algorithms that may advance the state-of-the-art in optimization and machine learning in general.

## 2 Preliminaries

In this section we introduce notations for linear transformers, data, and type of problems we consider.

### 2.1 Linear transformers and in-context learning

Given input sequence $e_1, e_2, ..., e_n \in \mathbb{R}^{d+1}$, a single head in a linear self-attention layer is usually parameterized by four matrices, key $W_K$, query $W_Q$, value $W_V$ and projection $W_P$. The output of the non-causal layer at position $i$ is $e_i + \Delta e_i$ where $\Delta e_i$ is computed as

$$\Delta e_i = W_P \left( \sum_{j=1}^n \langle W_Q e_i, W_K e_j \rangle W_V e_j \right). \tag{1}$$

Equivalently, one can use parameters $P = W_P W_V$ and $Q = W_K^\top W_Q$, and the equation becomes

$$\Delta e_i = \sum_{j=1}^n (e_j^\top Q e_i) P e_j. \tag{2}$$

Multiple heads $(P_1, Q_1), (P_2, Q_2), ..., (P_h, Q_h)$ simply sum their effects

$$\Delta e_i = \sum_{k=1}^H \sum_{j=1}^n (e_j^\top Q_k e_i) P_k e_j. \tag{3}$$

We define a *linear transformer* as a multi-layer neural network composed of $L$ linear self-attention layers parameterized by $\theta = \{Q_k^l, P_k^l\}$ for $k = 1 \dots H, l = 1 \dots L$. To isolate the core mechanisms, we consider a simplified decoder-only architecture, excluding MLPs and LayerNorm components. This architecture was also used in previous work (von Oswald et al., 2023a; Ahn et al., 2023).

We consider two versions of linear transformers: FULL with the transformer parameters represented by full matrices and DIAG, where the parameters are restricted to diagonal matrices only.

Inspired by von Oswald et al. (2023a), in this paper we consider regression data as the token sequence. Each token $e_i = (x_i, y_i) \in \mathbb{R}^{d+1}$ consists of a feature vector $x_i \in \mathbb{R}^d$ and its corresponding output

---

[1]We consider a model where each sequence contains data with the same noise level, while different sequences have different noise levels.

$y_i \in \mathbb{R}$. Additionally, we append a query token $e_{n+1} = (x_t, 0)$ to the sequence, where $x_t \in \mathbb{R}^d$ represents test data. The goal of in-context learning is to predict $y_t$ for the test data $x_t$. We constrain the attention to only focus on the first $n$ tokens of the sequence so that it ignores the query token.

We use $(x_i^l, y_i^l)$ to denote the $i$-th token in the transformer's output at layer $l$. The initial layer is simply the input: $(x_i^0, y_i^0) = (x_i, y_i)$. For a model with parameters $\theta$, we read out the prediction by taking the negative[2] of the last coordinate of the final token in the last layer as $\hat{y}_\theta(\{e_1, ..., e_n\}, e_{n+1}) = -y_{n+1}^L$.

Let's also define the following notation to be used throughout the paper

$$\Sigma = \sum_{i=1}^{n} x_i(x_i)^\top; \quad \alpha = \sum_{i=1}^{n} y_i x_i; \qquad\qquad \lambda = \sum_{i=1}^{n} (y_i)^2$$

$$\Sigma^l = \sum_{i=1}^{n} x_i^l(x_i^l)^\top; \quad \alpha^l = \sum_{i=1}^{n} y_i^l x_i^l; \qquad\qquad \lambda^l = \sum_{i=1}^{n} (y_i^l)^2$$

## 2.2 Noisy regression model

As a model problem, we consider data generated from a noisy linear regression model. For each input sequence $\tau$, we sample a ground-truth weight vector $w_\tau \sim N(0, I)$, and generate $n$ data points as $x_i \sim N(0, I)$ and $y_i = \langle w_\tau, x_i \rangle + \xi_i$, with noise $\xi_i \sim N(0, \sigma_\tau^2)$.

Note that each sequence can have different ground-truth weight vectors $w_\tau$, but every data point in the sequence shares the same $w_\tau$ and $\sigma_\tau$. The query is generated as $x_t \sim N(0, I)$ and $y_t = \langle w_\tau, x_t \rangle$ (since the noise is independent, whether we include noise in $y_q$ will only be an additive constant to the final objective).

We further define an ordinary least square (OLS) loss as

$$L_{\text{OLS}}(w) = \sum_{i=1}^{n} \left( y_i - \langle w, x_i \rangle \right)^2. \tag{4}$$

The OLS solution is $w^* := \Sigma^{-1}\alpha$ with residuals $r_i := y_i - \langle w^*, x_i \rangle$.

In the presence of noise $\sigma_\tau$, $w^*$ in general is not equal to the ground truth $w_\tau$. For a *known* noise level $\sigma_\tau$, the best estimator for $w_\tau$ is provided by ridge regression:

$$L_{\text{RR}}(w) = \sum_{i=1}^{n} \left( y_i - \langle w, x_i \rangle \right)^2 + \sigma_\tau^2 \|w\|^2, \tag{5}$$

with solution $w_{\sigma^2}^* := \left( \Sigma + \sigma_\tau^2 I \right)^{-1} \alpha$. Of course, in reality the variance of the noise is not known and has to be estimated from the data.

## 2.3 Fixed vs. mixed noise variance problems

We consider two different problems within the noisy linear regression framework.

**Fixed noise variance.** In this scenario, the variance $\sigma_\tau$ remains constant for all the training data. Here, the in-context loss is:

$$L(\theta) = \mathop{\mathbb{E}}_{\substack{w_\tau \sim N(0,I) \\ x_i \sim N(0,I) \\ \xi_i \sim N(0,\sigma_\tau^2)}} \left[ (\hat{y}_\theta(\{e_1, ..., e_n\}, e_{n+1}) - y_t)^2 \right], \tag{6}$$

where $e_i = (x_i, y_i)$ and $y_i = \langle w_\tau, x_i \rangle + \xi_i$. This problem was initially explored by Garg et al. (2022). Later, von Oswald et al. (2023a) have demonstrated that a linear transformer (6) converges to a form of a gradient descent solution, which they called GD$^{++}$. We define this in details later.

**Mixed noise variance.** In this case, the noise variance $\sigma_\tau$ is drawn from some fixed distribution $p(\sigma_\tau)$ for each sequence. The in-context learning loss becomes:

$$L(\theta) = \mathop{\mathbb{E}}_{\substack{w_\tau \sim N(0,I) \\ x_i \sim N(0,I) \\ \xi_i \sim N(0,\sigma_\tau^2) \\ \sigma_\tau \sim p(\sigma_\tau)}} \left[ (\hat{y}_\theta(\{e_1, ..., e_n\}, e_{n+1}) - y_t)^2 \right]. \tag{7}$$

---

[2]We set the actual prediction to $-y_{n+1}^l$, similar to von Oswald et al. (2023a), because it's easier for linear transformers to predict $-y_t$.

In other words, each training sequence $\tau$ has a fixed noise level $\sigma_\tau$, but different training sequences have different noise levels sampled from a specified distribution $p(\sigma_\tau)$. This scenario adds complexity because the model must predict $w_\tau$ for changing noise distribution, and the optimal solution likely would involve some sort of noise estimation. We have found that empirically, $GD^{++}$ fails to model this noise variance and instead converges to a solution which can be interpreted as a single noise variance estimate across all input data.

## 3 Related work

**In-context Learning as Gradient Descent**   Our work builds on research that frames in-context learning as (variants of) gradient descent (Akyürek et al., 2022; von Oswald et al., 2023a). For 1-layer linear transformer, several works Zhang et al. (2023); Mahankali et al. (2023); Ahn et al. (2023) characterized the optimal parameters and training dynamics. More recent works extended the ideas to auto-regressive models (Li et al., 2023; von Oswald et al., 2023b) and nonlinear models (Cheng et al., 2023). Fu et al. (2023) noticed that transformers perform similarly to second-order Newton methods on linear data, for which we give a plausible explanation in Theorem 5.1.

**In-context Learning in LLMs**   There are also many works that study how in-context learning works in pre-trained LLMs (Kossen et al., 2023; Wei et al., 2023; Hendel et al., 2023; Shen et al., 2023). Due to the complexity of such models, the exact mechanism for in-context learning is still a major open problem. Several works (Olsson et al., 2022; Chan et al., 2022; Akyürek et al., 2024) identified induction heads as a crucial mechanism for simple in-context learning tasks, such as copying, token translation and pattern matching.

**Other theories for training transformers**   Other than the setting of linear models, several other works (Garg et al., 2022; Tarzanagh et al., 2023; Li et al., 2023; Huang et al., 2023; Tian et al., 2023a,b) considered optimization of transformers under different data and model assumptions. Wen et al. (2023) showed that it can be difficult to interpret the "algorithm" performed by transformers without very strong restrictions.

**Mixed Linear Models**   Several works observed that transformers can achieve good performance on a mixture of linear models (Bai et al., 2023; Pathak et al., 2023; Yadlowsky et al., 2023). While these works show that transformers *c*an implement many variants of model-selection techniques, our result shows that linear transformers solve such problems by discovering interesting optimization algorithm with many hyperparameters tuned during the training process. Such a strategy is quite different from traditional ways of doing model selection. Transformers are also known to be able to implement strong algorithms in many different setups (Guo et al., 2023; Giannou et al., 2023).

**Effectiveness of linear and kernel-like transformers**   A main constraint on transformer architecture is that it takes $O(N^2)$ time for a sequence of length $N$, while for a linear transformer this can be improved to $O(N)$. Mirchandani et al. (2023) showed that even linear transformers are quite powerful for many tasks. Other works (Katharopoulos et al., 2020; Wang et al., 2020; Schlag et al., 2021; Choromanski et al., 2020) uses ideas similar to kernel/random features to improve the running time to almost linear while not losing much performance.

## 4 Linear transformers maintain linear regression model at every layer

While large, nonlinear transformers can model complex relationship, we show that linear transformers are restricted to maintaining a linear regression model based on the input, in the sense that the $l$-th layer output is always a linear function of the input with latent (and possibly nonlinear) coefficients.

**Theorem 4.1.** *Suppose the output of a linear transformer at $l$-th layer is $(x_1^l, y_1^l), (x_2^l, y_2^l), ..., (x_n^l, y_n^l), (x_t^l, y_t^l)$, then there exists matrices $M^l$, vectors $u^l, w^l$ and scalars $a^l$ such that*

$$x_i^{l+1} = M^l x_i + y_i u^l, \qquad\qquad x_t^{l+1} = M^l x_t,$$
$$y_i^{l+1} = a^l y_i - \langle w^l, x_i \rangle, \qquad\qquad y_t^{l+1} = -\langle w^l, x_t \rangle.$$

Note that $M^l, u^l, w^l$ and $a^l$ are not linear in the input, but this still poses restrictions on what the linear transformers can do. For example we show that it cannot represent a quadratic function:

**Theorem 4.2.** *Suppose the input to a linear transformer is $(x_1, y_1), (x_2, y_2), ..., (x_n, y_n)$ where $x_i \sim N(0, I)$ and $y_i = w^\top x_i$, let the $l$-th layer output be $(x_1^l, y_1^l), (x_2^l, y_2^l), ..., (x_n^l, y_n^l)$ and let $y^l = (y_1^l, ..., y_n^l)$ and $y^* = (x_1(1)^2, x_2(1)^2, ..., x_n(1)^2)$ (here $x_i(1)$ is just the first coordinate of $x_i$), then when $n \gg d$ with high probability the cosine similarity of $y^*$ and $y^l$ is at most 0.1.*

Theorem 4.1 implies that the output of linear transformer can always be explained as linear combinations of input with latent weights $a^l$ and $w^l$. The matrices $M^l$, vectors $u^l, w^l$ and numbers $a^l$ are not linear and can in fact be quite complex, which we characterize below:

**Lemma 4.3.** *In the setup of Theorem 4.1, if we let*

$$\begin{pmatrix} A^l & b^l \\ (c^l)^\top & d^l \end{pmatrix} :=$$

$$\sum_{k=1}^{h} \left[ P_k^l \sum_{j=1}^{n} \left( \begin{pmatrix} x_j^l \\ y_j^l \end{pmatrix} ((x_j^l)^\top, y_j^l) \right) Q_k^l \right],$$

*then one can recursively compute matrices $M^l$, vectors $u^l, w^l$ and numbers $a^l$ for every layer using*

$$M^{l+1} = (I + A^l)M^l + b^l(w^l)^\top$$
$$u^{l+1} = (I + A^l)u^l + a^l b^l$$
$$a^{l+1} = (1 + d^l)a^l + \langle c^l, u^l \rangle$$
$$w^{l+1} = (1 + d^l)w^l - (M^l)^\top c^l,$$

*with the init. condition $a^0 = 1, w^0 = 0, M^0 = I, u^0 = 0$.*

The updates to the parameters are complicated and nonlinear, allowing linear transformers to implement powerful algorithms, as we will later see in Section 5. In fact, even with diagonal $P$ and $Q$, they remain flexible. The updates in this case can be further simplified to a more familiar form:

**Lemma 4.4.** *In the setup of Theorem 4.1 with diagonal parameters, $u^l, w^l$ are updated as*

$$u^{l+1} = (I - \Lambda^l)u^l + \Gamma^l \Sigma \left( a^l w^* - w^l \right);$$
$$w^{l+1} = (1 + s^l)w^l - \Pi^l \Sigma (a^l w^* - w^l) - \Phi^l u^l.$$

*Here $\Lambda^l, \Gamma^l, s^l, \Pi^l, \Phi^l$ are matrices and numbers that depend on $M^l, u^l, a^l, w^l$ in Lemma 4.3.*

Note that $\Sigma \left( a^l w^* - w^l \right)$ is (proportional to) the gradient of a linear model $f(w^l) = \sum_{i=1}^{n}(a^l y_i - \langle w^l, x_i \rangle)^2$. This makes the updates similar to a gradient descent with momentum:

$$u^{l+1} = (1 - \beta)u^l + \nabla f(w^l); w^{l+1} = w^l - \eta u^l.$$

Of course, the formula in Lemma 4.4 is still much more complicated with matrices in places of $\beta$ and $\eta$, and also including a gradient term for the update of $w$.

# 5 Power of diagonal attention matrices

Although linear transformers are constrained, they can solve complex in-context learning problems. Empirically, we have found that they are able to very accurately solve linear regression with mixed noise variance (7), with final learned weights that are very diagonal heavy with some low-rank component (see Fig. 4). Surprisingly, the final loss remains remarkably consistent even when their $Q$ and $P$ matrices (3) are diagonal. Here we will analyze this special case and explain its effectiveness.

Since the elements of $x$ are permutation invariant, a diagonal parameterization reduces each attention heads to just four parameters:

$$P_k^l = \begin{pmatrix} p_{x,k}^l I & 0 \\ 0 & p_{y,k}^l \end{pmatrix}; \quad Q_k^l = \begin{pmatrix} q_{x,k}^l I & 0 \\ 0 & q_{y,k}^l \end{pmatrix}. \tag{8}$$

It would be useful to further reparametrize the linear transformer (3) using:

$$\begin{aligned} \omega_{xx}^l &= \sum_{k=1}^H p_{x,k}^l q_{x,k}^l, \quad \omega_{xy}^l = \sum_{k=1}^H p_{x,k}^l q_{y,k}^l, \\ \omega_{yx}^l &= \sum_{k=1}^H p_{y,k}^l q_{x,k}^l, \quad \omega_{yy}^l = \sum_{k=1}^H p_{y,k}^l q_{y,k}^l. \end{aligned} \tag{9}$$

This leads to the following diagonal layer updates:

$$\begin{aligned} x_i^{l+1} &= x_i^l + \omega_{xx}^l \Sigma^l x_i^l + w_{xy}^l y_i^l \alpha^l \\ x_t^{l+1} &= x_t^l + \omega_{xx}^l \Sigma^l x_t^l + w_{xy}^l y_t^l \alpha^l \\ y_i^{l+1} &= y_i^l + \omega_{yx}^l \langle \alpha^l, x_i^l \rangle + \omega_{yy}^l y_i^l \lambda^l, \\ y_t^{l+1} &= y_t^l + \omega_{yx}^l \langle \alpha^l, x_t^l \rangle + \omega_{yy}^l y_t^l \lambda^l. \end{aligned} \tag{10}$$

Four variables $\omega_{xx}^l, \omega_{xy}^l, \omega_{yx}^l, \omega_{yy}^l$ represent information flow between the data and the labels across layers. For instance, the term controlled by $\omega_{xx}^l$ measures information flow from $x^l$ to $x^{l+1}$, $\omega_{yx}^l$ measures the flow from $x^l$ to $y^{l+1}$ and so forth. Since the model can always be captured by these 4 variables, having many heads does not significantly increase its representation power. When there is only one head the equation $\omega_{xx}^l \omega_{yy}^l = \omega_{xy}^l \omega_{yx}^l$ is always true, while models with more than one head do not have this limitation. However empirically even models with one head is quite powerful.

### 5.1 GD$^{++}$ and least squares solver

GD$^{++}$, introduced in von Oswald et al. (2023a), represents a linear transformer that is trained on a fixed noise variance problem (6). It is a variant of a diagonal linear transformer, with all the heads satisfying $q_{y,k}^l = 0$. Dynamics are influenced only by $\omega_{xx}^l$ and $\omega_{yx}^l$, leading to simpler updates:

$$\begin{aligned} x_i^{l+1} &= \left( I + \omega_{xx}^l \Sigma^l \right) x_i^l \\ y_i^{l+1} &= y_i^l + \omega_{yx}^l \langle \alpha^l, x_i^l \rangle. \end{aligned} \tag{11}$$

The update on $x$ acts as preconditioning and the update on $y$ performs gradient descent on the data.

While existing analysis by Ahn et al. (2023) has not yielded fast convergence rates for GD$^{++}$, we show here that it is actually a second-order optimization algorithm for the least squares problem (4):

**Theorem 5.1.** *Given $(x_1, y_1), ..., (x_n, y_n), (x_t, 0)$ where $\Sigma$ has eigenvalues in the range $[\nu, \mu]$ with a condition number $\kappa = \nu/\mu$. Let $w^*$ be the optimal solution to least squares problem (4), then there exists hyperparameters for GD$^{++}$ algorithm that outputs $\hat{y}$ with accuracy $|\hat{y} - \langle x_t, w^* \rangle| \leq \epsilon \|x_t\| \|w^*\|$ in $l = O(\log \kappa + \log \log 1/\epsilon)$ steps. In particular that implies there exists an $l$-layer linear transformer that can solve this task.*

The convergence rate of $O(\log \log 1/\epsilon)$ is typically achieved only by second-order algorithms such as Newton's method.

### 5.2 Understanding $\omega_{yy}$: adaptive rescaling

If a layer only has $\omega_{yy}^l \neq 0$, it has a rescaling effect. The amount of scaling is related to the amount of noise added in a model selection setting. The update rule for this layer is:

$$y_i^{l+1} = \left( 1 + \omega_{yy}^l \lambda^l \right) y_i^l.$$

This rescales every $y$ by a factor that depends on $\lambda^l$. When $\omega_{yy}^l < 0$, this shrinks of the output based on the norm of $y$ in the previous layer. This is useful for the mixed noise variance problem, as ridge regression solution scales the least squares solution by a factor that depends on the noise level.

Specifically, assuming $\Sigma \approx \mathbb{E}[\Sigma] = nI$, the ridge regression solution becomes $w_{\sigma^2}^* \approx \frac{n}{n+\sigma^2} w^*$, which is exactly a scaled version of the OLS solution. Further, when noise is larger, the scaled factor is smaller, which agrees with the behavior of a negative $\omega_{yy}$.

We can show that using adaptive scaling $\omega_{yy}$ even a 2-layer linear transformer can be enough to solve a simple example of categorical mixed noise variance problem $\sigma_\tau \in \{\sigma_1, \sigma_2\}$ and $n \to \infty$:

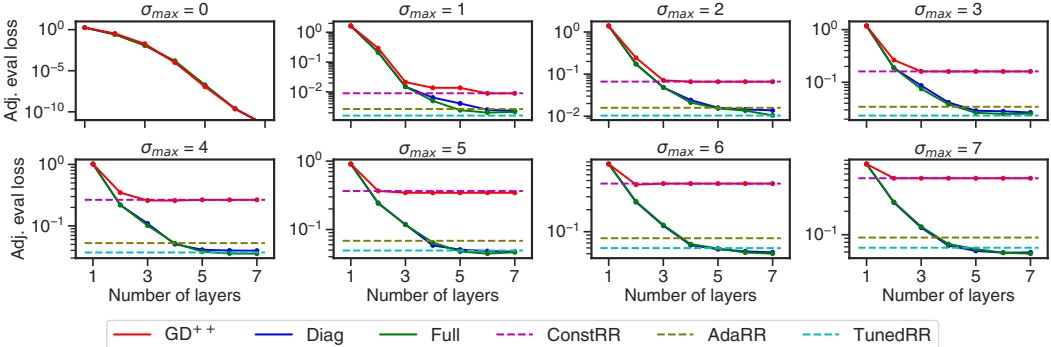

Figure 1: In-context learning performance for noisy linear regression problem across models with different number of layers and $\sigma_{max}$ for $\sigma_\tau \sim U(0, \sigma_{max})$. Each marker corresponds to a separately trained model with a given number of layers. Models with diagonal attention weights (DIAG) match those with full attention weights (FULL). Models specialized on a fixed noise (GD$^{++}$) perform poorly, similar to a Ridge Regression solution with a constant noise (CONSTRR). Among the baselines, only tuned exact Ridge Regression solution (TUNEDRR) is comparable with linear transformers.

**Theorem 5.2.** *Suppose the input to the transformer is $(x_1, y_1), (x_2, y_2), ..., (x_n, y_n), (x_q, 0)$, where $x_i \sim N(0, \frac{1}{n}I)$, $y_i = w^\top x_i + \xi_i$. Here $\xi_i \sim N(0, \sigma^2)$ is the noise whose noise level $\sigma$ can take one of two values: $\sigma_1$ or $\sigma_2$. Then as $n$ goes to $+\infty$, there exists a set of parameters for two-layer linear transformers such that the implicit $w^2$ of the linear transformer converges to the optimal ridge regression results (and the output of the linear transformer is $-\langle w^2, x_q \rangle$). Further, the first layer only has $\omega_{yx}$ being nonzero and the second layer only has $\omega_{yy}$ being nonzero.*

### 5.3 Understanding $\omega_{xy}$: adapting step-sizes

The final term in the diagonal model, $\omega_{xy}$, has a more complicated effect. Since it changes only the $x$-coordinates, it does not have an immediate effect on $y$. To understand how it influences the $y$ we consider a simplified two-step process, where the first step only has $\omega_{xy} \neq 0$ and the second step only has $\omega_{yx} \neq 0$ (so the second step is just doing one step of gradient descent). In this case, the first layer will update the $x_i$'s as:

$$x_i^1 = x_i + y_i \omega_{xy} \sum_{j=1}^n y_j x_j$$

$$= x_i + \omega_{xy} y_i \sum_{j=1}^n (\langle w^*, x_j \rangle + r_j) x_j$$

$$= x_i + \omega_{xy} y_i \Sigma w^*$$
$$= x_i + \omega_{xy} (\langle w^*, x_i \rangle + r_i) \Sigma w^*$$
$$= (I + \omega_{xy} \Sigma w^* (w^*)^\top) x_i + \omega_{xy} r_i \Sigma w^*.$$

There are two effects of the $\omega_{xy}$ term, one is a multiplicative effect on $x_i$, and the other is an additive term that makes $x$-output related to the residual $r_i$. The multiplicative step in $x_i$ has an unknown preconditioning effect. For simplicity we assume the multiplicative term is small, that is:

$$x_i^1 \approx x_i + \omega_{xy} r_i \Sigma w^*; \quad x_t^1 \approx x_t.$$

The first layer does not change $y$, so $y_t^1 = y_t$ and $y_i^1 = y_i$. For this set of $x_i$, we can write down the output on $y$ in the second layer as

$$y_t^2 = y_t + \omega_{yx} \sum_{i=1}^n y_i (x_i^1)^\top x_t$$

$$\approx y_t + \omega_{yx} [\sum_{i=1}^n y_i x_i + \omega_{xy} \sum_{i=1}^n y_i r_i \Sigma w^*] x_t$$

$$= y_t + \omega_{yx} (1 + \omega_{xy} \sum_{i=1}^n r_i^2)(\Sigma w^*)^\top x_t.$$

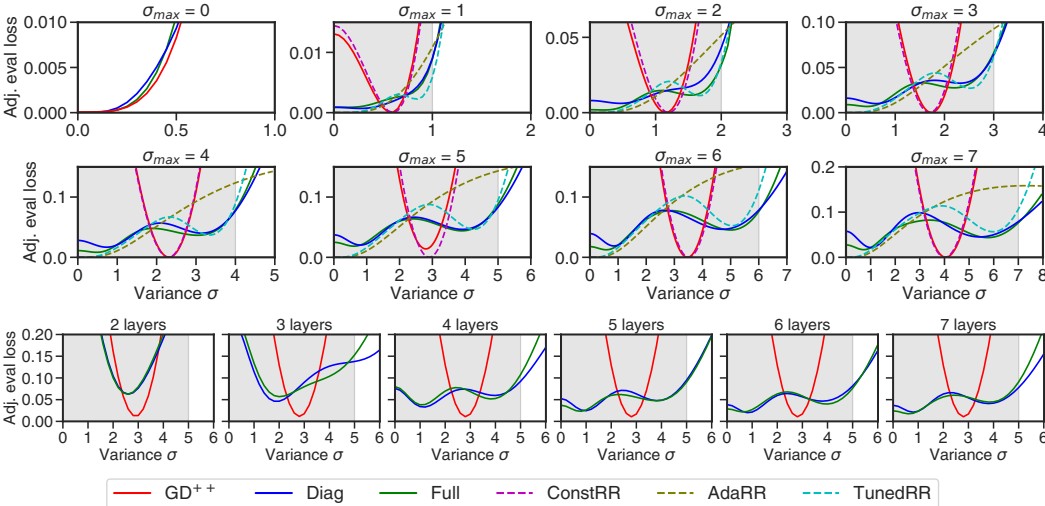

Figure 2: Per-variance profile of models behavior for uniform noise variance $\sigma_\tau \sim U(0, \sigma_{max})$. *Top two rows:* 7-layer models with varying $\sigma_{max}$. *Bottom row:* models with varying numbers of layers, fixed $\sigma_{max} = 5$. In-distribution noise is shaded gray.

Here we used the properties of residual $r_i$ (in particular $\sum_i y_i x_i = \Sigma w^*$, and $\sum_i y_i r_i = \sum_i r_i^2$). Note that $(\Sigma w^*)^\top x_t$ is basically what a gradient descent step on the original input should do. Therefore effectively, the two-layer network is doing gradient descent, but the step size is the product of $-\omega_{yx}$ and $(1 + \omega_{xy} \sum_i r_i^2)$. The factor $(1 + \omega_{xy} \sum_i r_i^2)$ depends on the level of noise, and when $\omega_{xy}, \omega_{yx} < 0$, the effective step size is smaller when there is more noise. This is especially helpful in the model selection problem, because intuitively one would like to perform early-stopping (small step sizes) when the noise is high.

## 6 Experiments

In this section, we investigate the training dynamics of linear transformers when trained with a mixed noise variance problem (7). We evaluate three types of single-head linear transformer models:

- FULL. Trains full parameter matrices.
- DIAG. Trains diagonal parameter matrices (10).
- GD$^{++}$. An even more restricted diagonal variant defined in (11).

For each experiment, we train each linear transformer modifications with a varying number of layers (1 to 7) using using Adam optimizer for $200\,000$ iterations with a learning rate of $0.0001$ and a batch size of $2\,048$. In some cases, especially for a large number of layers, we had to adjust the learning rate to prevent stability issues. We report the best result out of 5 runs with different training seeds. We used $N = 20$ in-context examples in $D = 10$ dimensions. We evaluated the algorithm using $100\,000$ novel sequences. All the experiments were done on a single H100 GPU with 80GB of VRAM. It took on average 4–12 hours to train a single algorithm, however experimenting with different weight decay parameters, better optimizer and learning rate schedule will likely reduce this number dramatically.

We use *adjusted evaluation loss* as our main performance metric. It is calculated by subtracting the oracle loss from the predictor's loss. The oracle loss is the closed-form solution of the ridge regression loss (5), assuming the noise variance $\sigma_\tau$ is known. The adjusted evaluation loss allows for direct model performance comparison across different noise variances. This is important because higher noise significantly degrades the model prediction. Our adjustment does not affect the model's optimization process, since it only modifies the loss by an additive constant.

**Baseline estimates.** We evaluated the linear transformer against a closed-form solution to the ridge regression problem (5). We estimated the noise variance $\sigma_\tau$ using the following methods:

- *Constant Ridge Regression (*CONSTRR*).* The noise variance is estimated using a single scalar value for all the sequences, tuned separately for each mixed variance problem.

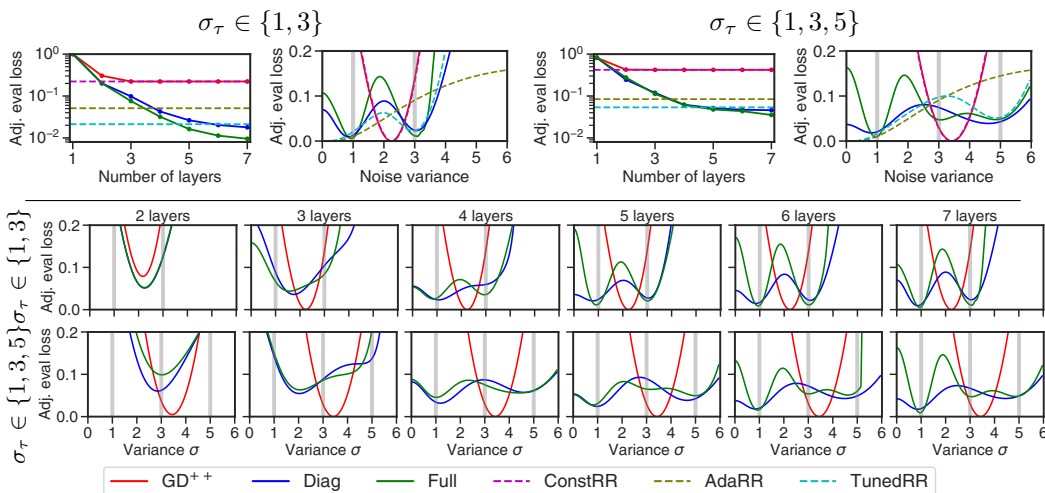

Figure 3: In-context learning performance for noisy linear regression across models with varying number of layers for conditional noise variance $\sigma_\tau \in \{1,3\}$ and $\sigma_\tau \in \{1,3,5\}$. *Top:* loss for models with various number of layers and per-variance profile for models with 7 layers. *Bottom:* Per-variance profile of the model across different numbers of layers. In-distribution noise is shaded gray.

- *Adaptive Ridge Regression (*ADARR*).* Estimate the noise variance via unbiased estimator (Cherkassky & Ma, 2003) $\sigma_{est}^2 = \frac{1}{n-d} \sum_{j=1}^n (y_j - \hat{y}_j)^2$, where $\hat{y}_j$ represents the solution to the ordinary least squares (4), found in a closed-form.
- *Tuned Adaptive Ridge Regression (*TUNEDRR*).* Same as above, but after the noise is estimated, we tuned two additional parameters to minimize the evaluation loss: (1) a max. threshold value for the estimated variance, (2) a multiplicative adjustment to the noise estimator. These values are tuned separately for each problem.

Notice that all the baselines above are based on ridge regression, which is a closed-form, non-iterative solution. Thus, they have an algorithmic advantage over linear transformers that do not have access to matrix inversion. These baselines help us gauge the best possible performance, establishing an upper bound rather than a strictly equivalent comparison.

A more faithful comparison to our method would be an iterative version of the ADARR that does not use matrix inversion. Instead, we can use gradient descent to estimate the noise and the solution to the ridge regression. However, in practice, this gradient descent estimator converges to ADARR only after $\approx 100$ iterations. In contrast, linear transformers typically converge in fewer than 10 layers.

We consider two choices for the distribution of $\sigma_\tau$:

- *Uniform.* $\sigma_\tau \sim U(0, \sigma_{max})$ drawn from a uniform distribution bounded by $\sigma_{max}$. We tried multiple scenarios with $\sigma_{max}$ ranging from 0 to 7.
- *Categorical.* $\sigma_\tau \in S$ chosen from a discrete set $S$. We tested $S = \{1,3\}$ and $S = \{1,3,5\}$.

Our approach generalizes the problem studied by Bai et al. (2023), who considered only categorical variance selection and show experiments only with two $\sigma_\tau$ values.

**Uniform noise variance.** For the uniform noise variance, Fig. 1 shows that FULL and DIAG achieve comparable performance across different numbers of layers and different $\sigma_{max}$. On the other hand, GD$^{++}$ converges to a higher value, closely approaching the performance of the CONSTRR baseline.

As $\sigma_{max}$ grows, linear transformers show a clear advantage over the baselines. With 4 layers, they outperform the closed-form solution ADARR for $\sigma_{max} = 4$ and larger. Models with 5 or more layers match or exceed the performance of TUNEDRR.

The top of Fig. 2 offers a detailed perspective on performance of 7-layer models and the baselines. Here, we computed per-variance profiles across noise variance range from 0 to $\sigma_{max} + 1$. We can see that poor performance of GD$^{++}$ comes from its inability to estimate well across the full noise variance range. Its performance closely mirrors to CONSTRR, suggesting that GD$^{++}$ under the hood might also be estimating a single constant variance for all the data.

ADARR perfectly estimates problems with no noise, but struggles more as noise variance increases. TUNEDRR slightly improves estimation by incorporating $\sigma_{max}$ into its tunable parameters, yet its

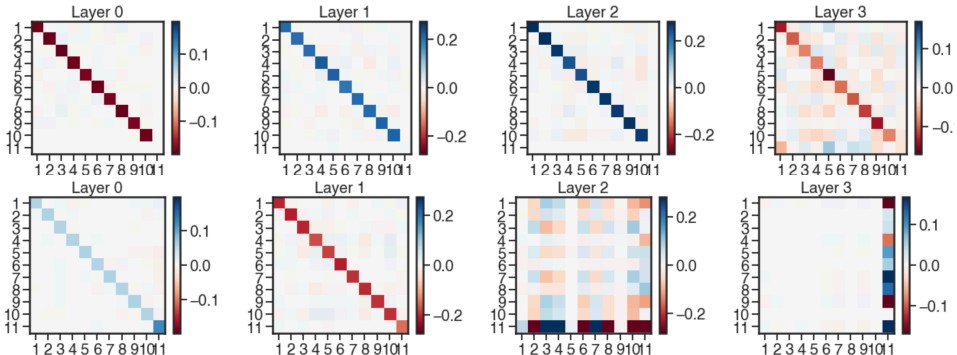

Figure 4: Weights for 4 layer linear transformer with FULL parametrization trained with categorical noise $\sigma_\tau \in \{1, 3\}$. *Top:* weights for $Q^l$ matrix, *bottom:* weights for $P^l$ matrix.

prediction suffers in the mid-range. FULL and DIAG demonstrate comparable performance across all noise variances. While more research is needed to definitively confirm or deny their equivalence, we believe that these models are actually not identical despite their similar performance.

At the bottom of Fig. 2 we set the noise variance to $\sigma_{max} = 5$ and display a per-variance profile for models with varying layers. Two-layer models for FULL and DIAG behave akin to GD$^{++}$, modeling only a single noise variance in the middle. However, the results quickly improve across the entire noise spectrum for 3 or more layers. In contrast, GD$^{++}$ quickly converges to a suboptimal solution.

**Categorical noise variance.** Fig. 3 shows a notable difference between DIAG and FULL models for categorical noise variance $\sigma_\tau \in \{1, 3\}$. This could stem from a bad local minima, or suggest a fundamental difference between the models for this problem. Interestingly, from per-variance profiling we see that DIAG extrapolates better for variances not used for training, while FULL, despite its lower in-distribution error, performs worse on unseen variances. Fig. 4 shows learned weights of the 4 layer linear transformer with FULL parametrization. The weights are very diagonal heavy, potentially with some low-rank component.

For $\sigma_\tau \in \{1, 3, 5\}$, examining the per-variance profile at the bottom of Fig. 3 reveals differences in their behaviors. FULL exhibits a more complex per-variance profile with more fluctuations than the diagonal model, suggesting greater representational capacity. Surprisingly, it did not translate to better loss results compared to DIAG.

For easy comparison, we compile the results of all methods and baselines in Table 1 in the Appendix.

# 7   Conclusions

Our research reveals the surprising ability of linear transformers to tackle challenging in-context learning problems. We show that each layer maintains an implicit linear regression model, akin to a complex variant of preconditioned gradient descent with momentum-like behavior.

Remarkably, when trained on noisy linear regression problems with unknown noise variance, linear transformers not only outperform standard baselines but also uncover a sophisticated optimization algorithm that incorporates noise-aware step-size adjustments and rescaling. This discovery highlights the potential of linear transformers to automatically discover novel optimization algorithms when presented with the right problems, opening exciting avenues for future research, including automated algorithm discovery using transformers and generalization to other problem domains.

While our findings demonstrate the impressive capabilities of linear transformers in learning optimization algorithms, we acknowledge limitations in our work. These include the focus on simplified linear models, analysis of primarily diagonal attention matrices, and the need for further exploration into the optimality of discovered algorithms, generalization to complex function classes, scalability with larger datasets, and applicability to more complex transformer architectures. We believe these limitations present valuable directions for future research and emphasize the need for a deeper understanding of the implicit learning mechanisms within transformer architectures.

# 8   Acknowledgements

The authors would like to thank Nolan Miller and Andrey Zhmoginov for their valuable suggestions and feedback throughout the development of this project. Part of this work was done while Rong Ge was visiting Google Research. Rong Ge's research is supported in part by NSF Award DMS-2031849 and CCF-1845171 (CAREER).

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

# A Proofs from Sections 4 and 5

## A.1 Proof of Theorem 4.1

We first give the proof for Theorem 4.1. In the process we will also prove Lemma 4.3, as Theorem 4.1 follows immediately from an induction based on the lemma.

*Proof.* We do this by induction. At $l = 0$, it's easy to check that we can set $a^{(0)} = 1, w^{(0)} = 0, M^{(0)} = I, u^{(0)} = 0$.

Suppose this is true for some layer $l$, if the weights of layer $l$ are $(P_1^l, Q_1^l), ..., (P_k^l, Q_k^l)$ for $k$ heads, at output of layer $l + 1$ we have:

$$\begin{pmatrix} x_i^{l+1} \\ y_i^{l+1} \end{pmatrix} = \begin{pmatrix} x_i^l \\ y_i^l \end{pmatrix} + \sum_{k=1}^{p} \left[ P_k^l \sum_{j=1}^{n} \left( \begin{pmatrix} x_j^l \\ y_j^l \end{pmatrix} ((x_j^l)^\top, y_j^l) \right) Q_k^l \right] \begin{pmatrix} x_i^l \\ y_i^l \end{pmatrix}. \tag{12}$$

Note that the same equation is true for $i = n + 1$ just by letting $y_{n+1} = 0$. Let the middle matrix has the following structure:

$$\begin{pmatrix} A & b \\ c^\top & d \end{pmatrix} := \sum_{k=1}^{p} \left[ P_k^l \sum_{j=1}^{n} \left( \begin{pmatrix} x_j^l \\ y_j^l \end{pmatrix} ((x_j^l)^\top, y_j^l) \right) Q_k^l \right],$$

Then one can choose the parameters of the next layer as in Lemma 4.3

$$M^{l+1} = (I + A)M^l + b(w^l)^\top$$
$$u^{l+1} = (I + A)u^l + a^l b$$
$$a^{l+1} = (1 + d)a^l + \langle c, u^l \rangle$$
$$w^{l+1} = (1 + d)w^l - (M^l)^\top c.$$

One can check that this choice satisfies (12). $\qquad\square$

## A.2 Proof of Lemma 4.4

This lemma is in fact a corollary of Lemma 4.3. We first give a more detailed version which explicitly state the unknown matrices $\Lambda^l, \Gamma^l, \Pi^l, \Phi^l$:

**Lemma A.1.** *In the setup of Theorem 4.1 with diagonal parameters* (9)*, one can recursively compute matrices $u^l, w^l$ using the following formula*

$$\begin{aligned} u^{l+1} &= \left( (1 + \omega_{xy}^l (a^l)^2 \rho) I + \omega_{xx}^l \Sigma^l \right) u^l \\ &\quad + a^l \omega_{xy}^l \left( M^l + a^l u^l (w^*)^\top \right) \Sigma \left( a^l w^* - w^l \right), \\ w^{l+1} &= (1 + \omega_{yy}^l \lambda^l) w^l \\ &\quad - \omega_{yx}^l (M^l)^\top (M^l + a^l u^l (w^*)^\top) \Sigma (a^l w^* - w^l) \\ &\quad - a^l \rho \omega_{yx}^l (M^l)^\top u^l, \end{aligned}$$

*where $\rho = \sum_{i=1}^{n} r_i^2$ and initial conditions $a^0 = 1, w^0 = 0, M^0 = I, u^0 = 0$*

*Proof.* First, we compute the following matrix that appeared in Lemma 4.3 for the specific diagonal case:

$$\begin{aligned} \begin{pmatrix} A^l & b^l \\ (c^l)^\top & d^l \end{pmatrix} &= \sum_{k=1}^{p} \left[ P_k^l \sum_{j=1}^{n} \left( \begin{pmatrix} x_j^l \\ y_j^l \end{pmatrix} ((x_j^l)^\top, y_j^l) \right) Q_k^l \right], \\ &= \begin{pmatrix} \omega_{xx}^l \Sigma^l & \omega_{xy}^l \alpha^l \\ \omega_{yx}^l (\alpha^l)^\top & \omega_{yy}^l \lambda^l \end{pmatrix}. \end{aligned}$$

This implies that $A^l = \omega_{xx}^l \Sigma^l$, $b^l = \omega_{xy}^l \alpha^l$, $c^l = \omega_{yx}^l \alpha^l$ and $d^l = \omega_{yy}^l \lambda^l$. Next we rewrite $\alpha^l$:

$$\alpha^l = \sum_{i=1}^n y^l x^l$$
$$= \sum_{i=1}^n (a^l y_i - \langle w^l, x_i \rangle)(M^l x_i + y_i u^l)$$
$$= \sum_{i=1}^n (a^l r_i + \langle a^l w^* - w^l, x_i \rangle)((M^l + a^l u^l (w^*)^\top) x_i + r_i u^l)$$
$$= \sum_{i=1}^n \langle a^l w^* - w^l, x_i \rangle (M^l + a^l u^l (w^*)^\top) x_i + \sum_{i=1}^n a^l r_i^2 u^l$$
$$= (M^l + a^l u^l (w^*)^\top) \Sigma (a^l w^* - w^l) + a^l \rho u^l.$$

Here the first step is by Theorem 4.1, the second step replaces $y_i$ with $\langle w^*, x_i \rangle + r_i$, the third step uses the fact that $\sum_{i=1}^n r_i x_i = 0$ to get rid of the cross terms.

The remaining proof just substitutes the formula for $\alpha^l$ into Lemma 4.3.

$\square$

Now Lemma A.1 implies Lemma 4.4 immediately by setting $\Lambda^l = -\omega_{xy}^l (a^l)^2 \rho I - \omega_{xx}^l \Sigma^l$, $\Gamma^l = a^l \omega_{xy}^l \left( M^l + a^l u^l (w^*)^\top \right)$, $s^l = tu\omega_{yy} \lambda^l$, $\Pi^l = \omega_{yx}^l (M^l)^\top (M^l + a^l u^l (w^*)^\top)$ and $\Phi^l = a^l \rho \omega_{yx}^l (M^l)^\top$.

### A.3 Proof for Theorem 4.2

*Proof.* By Theorem 4.1, we know $y_i^l = \langle w^l, x_i \rangle$ for some $w^l$. When $n \gg d$, with high probability the norm of $y^l$ is on the order of $\Theta(\sqrt{n}) \|w^l\|$, and the norm of $y^*$ is $\Theta(\sqrt{n})$. Therefore we only need to bound the correlation. The correlation is equal to

$$|\langle y^*, y^l \rangle| = |w_1^l \sum_{i=1}^n x_1^3 + \sum_{i=1}^n x_i(1)^2 \sum_{j=2}^d w_j^l x_i(j)|.$$

We know with high probability $|\sum_{i=1}^n x_i^3| = O(\sqrt{n})$ because $\mathbb{E}[x_i^3] = 0$. The second term can be written as $\langle w^l, v \rangle$ where $v$ is a vector whose coordinates are $v_1 = 0$ and $v_j = \sum_{i=1}^n x_i(1)^2 x_i(j)$ for $2 \le j \le d$, therefore with high probability $\|v\| = O(\sqrt{nd})$. Therefore, with high probability the cosine similarity is at most

$$\frac{|\langle y^*, y^l \rangle|}{\|y^*\| \|y^l\|} = O(1) \frac{|\langle y^*, y^l \rangle|}{n \|w^l}$$
$$= O(1) \frac{|w_1^l \sum_{i=1}^n x_1^3 + \sum_{i=1}^n x_i(1)^2 \sum_{j=2}^d w_j^l x_i(j)|}{n \|w^l}$$
$$\le O(1) \frac{|w_1^l| |\sum_{i=1}^n x_1^3| + |w^l| \|v\|}{n \|w^l}$$
$$\le O(1) \frac{\sqrt{n} + \sqrt{nd}}{n}.$$

When $n \gg d$ this can be made smaller than any fixed constant. $\square$

### A.4 Proof for Theorem 5.1

In this section we prove Theorem 5.1 by finding hyperparameters for GD$^{++}$ algorithm that solves least squares problems with very high accuracy. The first steps in the construction iteratively makes the data $x_i$'s better conditioned, and the last step is a single step of gradient descent. The proof is based on several lemma, first we observe that if the data is very well-conditioned, then one-step gradient descent solves the problem accurately:

**Lemma A.2.** *Given $(x_1, y_1), ..., (x_n, y_n)$ where $\Sigma := \sum_{i=1}^{n} x_i x_i^\top$ has eigenvalues between $1$ and $1 + \epsilon$. Let $w^* := \arg\min_w \sum_{i=1}^{n} (y_i - \langle w, x_i \rangle)^2$ be the optimal least squares solution, then $\hat{w} = \sum_{i=1}^{n} y_i x_i$ satisfies $\|\hat{w} - w^*\| \leq \epsilon \|w^*\|$.*

*Proof.* We can write $y_i = \langle x_i, w^* \rangle + r_i$. By the fact that $w^*$ is the optimal solution we know $r_i$'s satisfy $\sum_{i=1}^{n} r_i x_i = 0$. Therefore $\hat{w} = \sum_{i=1}^{n} y_i x_i = \sum_{i=1}^{n} \langle x_i, w^* \rangle x_i = \Sigma w^*$. This implies

$$\|\hat{w} - w^*\| = \|(\Sigma - I)w^*\| \leq \|\Sigma - I\|\|w^*\| \leq \epsilon \|w^*\|.$$

□

Next we show that by applying just the preconditioning step of GD$^{++}$, one can get a well-conditioned $x$ matrix very quickly. Note that the $\Sigma$ matrix is updated as $\Sigma \leftarrow (I - \gamma\Sigma)\Sigma(I - \gamma\Sigma)$, so an eigenvalue of $\lambda$ in the original $\Sigma$ matrix would become $\lambda(1 - \gamma\lambda)^2$. The following lemma shows that this transformation is effective in shrinking the condition number

**Lemma A.3.** *Suppose $\nu/\mu = \kappa \geq 1.1$, then there exists an universal constant $c < 1$ such that choosing $\gamma\nu = 1/3$ implies*

$$\frac{\max_{\lambda \in [\nu, \mu]} \lambda(1 - \gamma\lambda)^2}{\min_{\lambda \in [\nu, \mu]} \lambda(1 - \gamma\lambda)^2} \leq c\kappa.$$

*On the other hand, if $\nu/\mu = \kappa \leq 1 + \epsilon$ where $\epsilon \leq 0.1$, then choosing $\gamma\nu = 1/3$ implies*

$$\frac{\max_{\lambda \in [\nu, \mu]} \lambda(1 - \gamma\lambda)^2}{\min_{\lambda \in [\nu, \mu]} \lambda(1 - \gamma\lambda)^2} \leq 1 + 2\epsilon^2.$$

The first claim shows that one can reduce the condition number by a constant factor in every step until it's a small constant. The second claim shows that once the condition number is small $(1 + \epsilon)$, each iteration can bring it much closer to 1 (to the order of $1 + O(\epsilon^2)$).

Now we prove the lemma.

*Proof.* First, notice that the function $f(x) = x(1 - \gamma x)^2$ is monotonically nondecreasing for $x \in [0, \nu]$ if $\gamma\nu = 1/3$ (indeed, it's derivative $f'(x) = (1 - \gamma x)(1 - 3\gamma x)$ is always nonnegative). Therefore, the max is always achieved at $x = \nu$ and the min is always achieved at $x = \mu$. The new ratio is therefore

$$\frac{\nu(1 - \gamma\nu)^2}{\mu(1 - \gamma\mu)^2} = \kappa\frac{4/9}{(1 - 1/3\kappa)^2}.$$

When $\kappa \geq 1.1$ the ratio $\frac{4/9}{(1 - 1/3\kappa)^2}$ is always below $\frac{4/9}{(1 - 1/3.3)^2}$ which is a constant bounded away from 1.

When $\kappa = 1 + \epsilon < 1.1$, we can write down the RHS in terms of $\epsilon$

$$\frac{\nu(1 - \gamma\nu)^2}{\mu(1 - \gamma\mu)^2} = \kappa\frac{4/9}{(1 - 1/3\kappa)^2} = (1 + \epsilon)(1 + \frac{1}{2}(1 - \frac{1}{1 + \epsilon}))^{-2}.$$

Note that by the careful choice of $\gamma$, the RHS has the following Taylor expansion:

$$(1 + \epsilon)(1 + \frac{1}{2}(1 - \frac{1}{1 + \epsilon}))^{-2} = 1 + \frac{3\epsilon^2}{4} - \frac{5\epsilon^3}{4} + O(\epsilon^4).$$

One can then check the RHS is always upperbounded by $2\epsilon^2$ when $\epsilon < 0.1$.

□

With the two lemmas we are now ready to prove the main theorem:

*Proof.* By Lemma A.3 we know in $O(\log \kappa + \log\log 1/\epsilon)$ iterations, by assigning $\kappa$ in the way of Lemma A.3 one can reduce the condition number of $x$ to $\kappa' \leq 1 + \epsilon/2\kappa$ (we chose $\epsilon/2\kappa$ here to give some slack for later analysis).

Let $\Sigma'$ be the covariance matrix after these iterations, and $\nu', \mu'$ be the upper and lowerbound for its eigenvalues. The data $x_i$'s are transformed to a new data $x_i' = Mx_i$ for some matrix

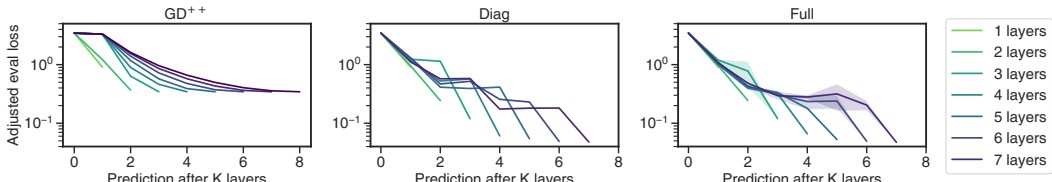

Figure 5: Linear transformer models show a consistent decrease in error per layer when trained on data with mixed noise variance $\sigma_\tau \sim U(0, 5)$. The error bars measure variance over 5 training seeds.

$M$. Let $M = A\Sigma^{-1/2}$, then since $M' = AA^\top$ we know $A$ is a matrix with singular values between $\sqrt{\mu'}$ and $\sqrt{\nu'}$. The optimal solution $(w^*)' = M^{-\top}w^*$ has norm at most $\sqrt{\nu}/\sqrt{\mu'}\|w^*\|$. Therefore by Lemma A.2 we know the one-step gradient step with $\hat{w} = \sum_{i=1}^n \frac{1}{\mu'} y_i x_i$ satisfy $\|\hat{w} - (w^*)'\| \leq (\kappa' - 1)\sqrt{\nu}/\sqrt{\mu'}\|w^*\|$. The test data $x_t$ is also transformed to $x_t' = A\Sigma^{-1/2}x_t$, and the algorithm outputs $\langle \hat{w}, x_t' \rangle$, so the error is at most $\sqrt{\nu}\|w^*\| * \|x_t'\| \leq (\kappa' - 1)\sqrt{\kappa}\sqrt{\kappa'}\|w^*\| * \|x_t\|$. By the choice of $\kappa'$ we can check that RHS is at most $\epsilon\|w^*\|\|x_t\|$. □

### A.5 Proof of the Theorem 5.2

*Proof.* The key observation here is that when $n \to \infty$, under the assumptions we have $\lim_{n\to\infty} \sum_{i=1}^n x_i x_i^\top = I$. Therefore the ridge regression solutions converge to $w_{\sigma^2}^* = \frac{1}{1+\sigma^2} \sum_{i=1}^n y_i x_i$ and the desired output is $\langle w_{\sigma^2}^*, x_q \rangle$.

By the calculations before, we know after the first-layer, the implicit $w$ is $w^1 = \omega_{yx} \sum_{i=1}^n y_i x_i$. As long as $\omega_{yx}$ is a constant, when $n \to \infty$ we know $\frac{1}{n} \sum_{i=1}^n (y_i^1)^2 = \sigma^2$ (as the part of $y$ that depend on $x$ is negligible compared to noise), therefore the output of the second layer satisfies

$$w^2 = (1 + n\sigma^2\omega_{yy})w^1 = (1 + n\sigma^2\omega_{yy})\omega_{yx} \sum_{i=1}^n y_i x_i.$$

Therefore, as long as we choose $\omega_{yx}$ and $\omega_{yy}$ to satisfy $(1 + n\sigma^2\omega_{yy})\omega_{yx} = \frac{1}{1+\sigma^2}$ when $\sigma = \sigma_1$ or $\sigma_2$ (notice that these are two linear equations on $\omega_{yx}$ and $n\omega_{yx}\omega_{yy}$, so they always have a solution), then we have $\lim_{n\to\infty} w^2 = w_{\sigma^2}^*$ for the two noise levels. □

## B    More experiments

Here we provide results of additional experiments that did not make it to the main text.

Fig. 6 shows an example of unadjusted loss. Clearly, it is virtually impossible to compare the methods across various noise levels this way.

Fig. 7 shows per-variance profile of intermediate predictions of the network of varying depth. It appears that GD$^{++}$ demonstrates behavior typical of GD-based algorithms: early iterations model higher noise (similar to early stopping), gradually converging towards lower noise predictions. DIAG exhibits this patter initially, but then dramatically improves, particularly for lower noise ranges. Intriguingly, FULL displays the opposite trend, first improving low-noise predictions, followed by a decline in higher noise prediction accuracy, especially in the last layer.

Finally, Table 1 presents comprehensive numerical results for our experiments across various mixed noise variance models. For each model variant (represented by a column), the best-performing result is highlighted in bold.

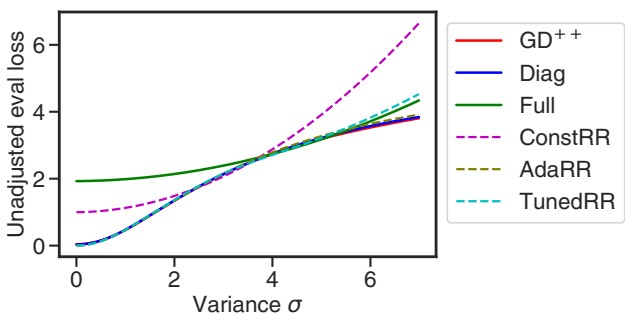

Figure 6: Example of unadjusted loss given by directly minimizing (7). It is pretty hard to see variation between comparable methods using this loss directly.

| Method | Uniform $\sigma_\tau \sim (0, \sigma_{max})$ | | | | | | | | Categorical $\sigma_\tau \in S$ | |
| --- | --- | --- | --- | --- | --- | --- | --- | --- | --- | --- |
| | 0 | 1 | 2 | 3 | 4 | 5 | 6 | 7 | {1,3} | {1,3,5} |
| 1 layer | | | | | | | | | | |
| GD$^{++}$ | 1.768 | 1.639 | 1.396 | 1.175 | 1.015 | 0.907 | 0.841 | 0.806 | 1.007 | 0.819 |
| DIAG | 1.767 | 1.639 | 1.396 | 1.175 | 1.015 | 0.906 | 0.841 | 0.806 | 1.007 | 0.819 |
| FULL | 1.768 | 1.640 | 1.397 | 1.176 | 1.016 | 0.907 | 0.842 | 0.806 | 1.008 | 0.820 |
| 2 layers | | | | | | | | | | |
| GD$^{++}$ | 0.341 | 0.295 | 0.243 | 0.265 | 0.347 | 0.366 | 0.440 | 0.530 | 0.305 | 0.427 |
| DIAG | 0.265 | 0.214 | 0.173 | 0.188 | 0.219 | 0.242 | 0.254 | 0.259 | 0.201 | 0.246 |
| FULL | 0.264 | 0.215 | 0.173 | 0.188 | 0.220 | 0.245 | 0.259 | 0.263 | 0.202 | 0.276 |
| 3 layers | | | | | | | | | | |
| GD$^{++}$ | 0.019 | 0.021 | 0.071 | 0.161 | 0.259 | 0.344 | 0.454 | 0.530 | 0.222 | 0.422 |
| DIAG | 0.013 | 0.015 | 0.048 | 0.087 | 0.109 | 0.118 | 0.121 | 0.123 | 0.098 | 0.119 |
| FULL | 0.012 | 0.015 | 0.049 | 0.075 | 0.101 | 0.117 | 0.124 | 0.127 | 0.076 | 0.113 |
| 4 layers | | | | | | | | | | |
| GD$^{++}$ | 9.91e-05 | 0.014 | 0.066 | 0.160 | 0.258 | 0.344 | 0.454 | 0.530 | 0.222 | 0.422 |
| DIAG | 1.19e-04 | 0.006 | 0.024 | 0.041 | 0.050 | 0.059 | 0.065 | 0.073 | 0.043 | 0.062 |
| FULL | 1.63e-04 | 0.005 | 0.021 | 0.038 | 0.052 | 0.065 | 0.068 | 0.076 | 0.032 | 0.061 |
| 5 layers | | | | | | | | | | |
| GD$^{++}$ | 1.14e-07 | 0.014 | 0.066 | 0.161 | 0.265 | 0.344 | 0.454 | 0.530 | 0.222 | 0.422 |
| DIAG | 1.81e-07 | 0.004 | 0.016 | 0.029 | 0.041 | 0.051 | 0.058 | 0.062 | 0.026 | 0.051 |
| FULL | 1.79e-07 | **0.002** | 0.015 | 0.026 | 0.038 | 0.048 | 0.059 | 0.065 | 0.016 | 0.048 |
| 6 layers | | | | | | | | | | |
| GD$^{++}$ | 2.37e-10 | 0.009 | 0.066 | 0.161 | 0.265 | 0.344 | 0.454 | 0.530 | 0.222 | 0.422 |
| DIAG | 2.57e-10 | 0.003 | 0.014 | 0.028 | 0.040 | 0.048 | 0.054 | 0.059 | 0.020 | 0.047 |
| FULL | 2.71e-10 | **0.002** | 0.014 | 0.025 | 0.036 | 0.044 | 0.052 | 0.059 | 0.011 | 0.043 |
| 7 layers | | | | | | | | | | |
| GD$^{++}$ | 2.65e-12 | 0.009 | 0.066 | 0.161 | 0.265 | 0.344 | 0.454 | 0.530 | 0.222 | 0.422 |
| DIAG | 2.50e-12 | **0.002** | 0.014 | 0.027 | 0.040 | 0.047 | 0.052 | 0.059 | 0.018 | 0.046 |
| FULL | 2.50e-12 | **0.002** | **0.010** | 0.025 | **0.035** | **0.047** | **0.050** | **0.057** | **0.010** | **0.035** |
| Baselines | | | | | | | | | | |
| CONSTRR | **0** | 0.009 | 0.066 | 0.161 | 0.265 | 0.365 | 0.454 | 0.530 | 0.222 | 0.422 |
| ADARR | **0** | 0.003 | 0.016 | 0.034 | 0.053 | 0.068 | 0.081 | 0.092 | 0.051 | 0.084 |
| TUNEDRR | **0** | **0.002** | **0.010** | **0.023** | 0.037 | 0.049 | 0.060 | 0.068 | 0.021 | 0.054 |

Table 1: Adjusted evaluation loss for models with various number of layers with uniform noise variance $\sigma_\tau \sim U(0, \sigma_{max})$. We highlight in bold the best results for each problem setup (i.e. each column).

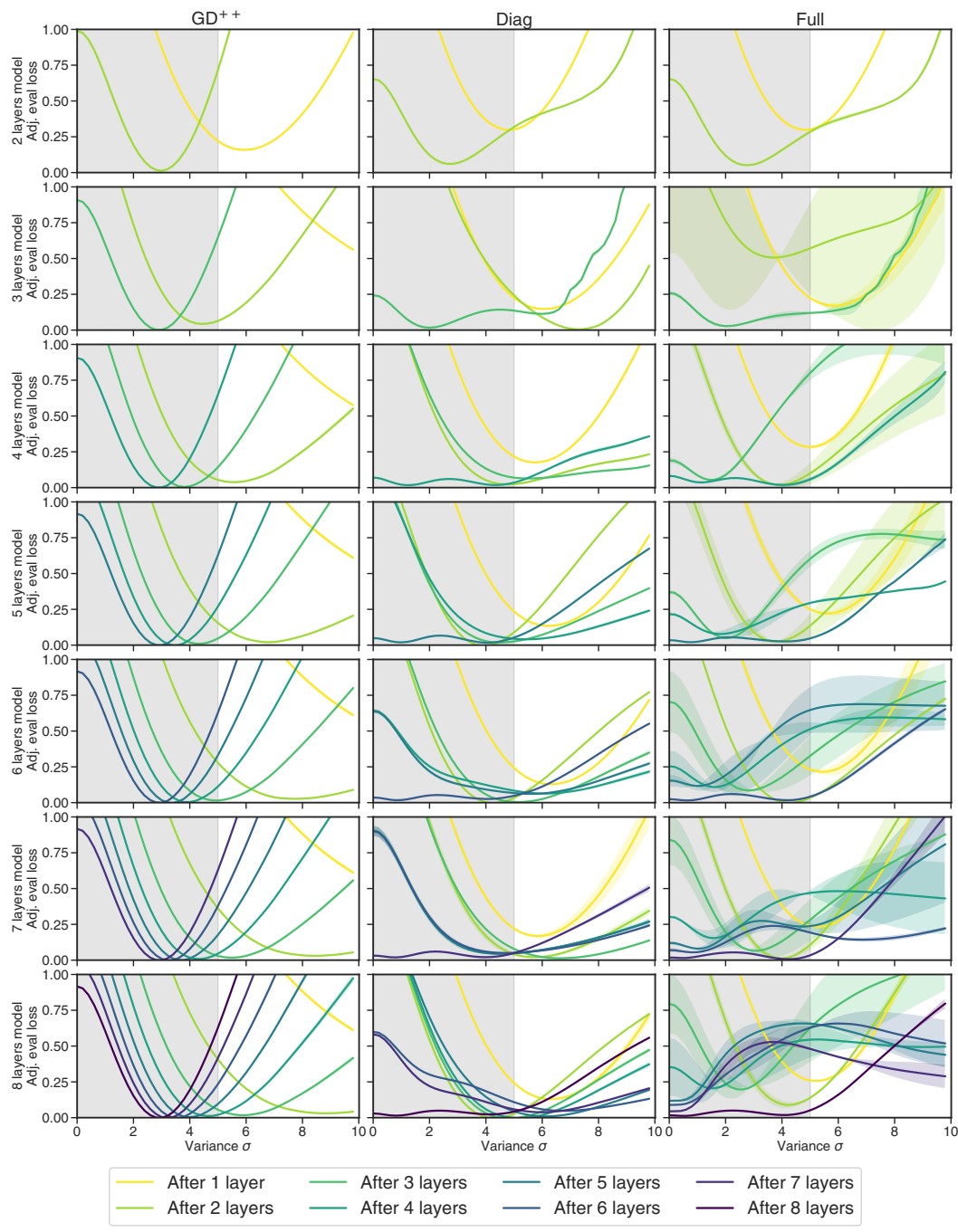

Figure 7: Layer by layer prediction quality for different models with $\sigma_\tau \sim U(0,5)$. The error bars measure std over $5$ training seeds.

