# OpenReview forum: "Linear Transformers are Versatile In-Context Learners"
_NeurIPS.cc/2024/Conference — NeurIPS 2024 poster_

### Official Review · Reviewer_dAYB · 2024-07-08

**Soundness:** 4
**Presentation:** 3
**Contribution:** 3
**Rating:** 6
**Confidence:** 4

**Summary:**

This paper proves that linear transformer layers maintain a weight vector for implicit linear regressions, including a more challenging scenario where data is corrupted with different levels of noise. In the theoretical analysis, this paper shows the intrinsic mechanism of the  gradient scent in linear attention, where implicit variants play various roles in parameter update. Besides, this paper shows that simplified linear attention with diagonal transformation also maintains a powerful GD behavior. The experiments demonstrate the flexibility of linear transformers which outperforms GD++ as well as other linear regression solutions. This research promotes the understanding of Transformer weights and implicit learning capabilities of attention-based models.

**Strengths:**

1. The theoretical analysis in this paper is sound and insightful for understanding the behavior of linear attention.
2. This paper broadens the analysis framework from fixed noise linear regression to flexible noise, making a step forward to more complex problems.
3. The experiment settings can prove the mathematical formulation in this paper.

**Weaknesses:**

The analysis framework is still on linear regression and linear attention with simplified modifications, which is similar to previous references. The theory can not be generalized to other architectures and context settings.

**Questions:**

1. How do you think about different linear attention variants in theory? For example, the most trivial one is $QK^TV$, but there are many improved versions, such as RetNet, SSM, and GLA. Do these architectures bring stronger modeling capability in theory?
2. With linear regression layer-wisely, does your claim that linear Transformer with FFN only maintains linear regression still work? For example, in Theorem 4.2, it is easy to build a quadratic FFN to make more complex computations.

**Limitations:**

The limitation is discussed in the conclusion section.

---

> ### Author Rebuttal · Authors · 2024-08-05
>
> We thank the reviewer for raising these insightful questions, which highlight the importance of generalization of our findings. While a complete exploration of these aspects is beyond the scope of our current work, we would like to provide some initial thoughts and discuss potential future directions.
>
> > How do you think about different linear attention variants in theory? For example, the most trivial one is $QK^TV$, but there are many improved versions, such as RetNet, SSM, and GLA. Do these architectures bring stronger modeling capability in theory?
>
> While we focused on simple attention for clarity and ease of analysis, these more advanced architectures could indeed offer stronger modeling capabilities in theory. They can implicitly capture higher-order feature interactions. These architectures often employ techniques like kernel functions or randomization, which could allow them to implicitly capture higher-order interactions and learn richer function classes. More specifically, we believe our Theorem 4.1/4.2 would generalize to the linear version of RetNet, but would not generalize to SSM and GLA (because SSM is time-varying and GLA has nonlinearities in the form of gates). Even in the setting of RetNet, the specific algorithms discovered could be more sophisticated. Exploring how our findings extend to these richer attention mechanisms is a promising direction for future work.
>
> > With linear regression layer-wisely, does your claim that linear Transformer with FFN only maintains linear regression still work? For example, in Theorem 4.2, it is easy to build a quadratic FFN to make more complex computations.
>
> Indeed, our current analysis considers linear transformers without FFNs. This simplification allows us to isolate the core behavior of the attention mechanism. With nonlinear FFNs, Theorem 4.2 no longer holds. A quadratic FFN can indeed introduce higher-order relationships, however the output of each attention layer is still a linear combination of its inputs (though the weights might be non-linear functions of the input). Exploring the complex interaction between attention and FFNs in the context of implicit optimization is crucial for future research and may require more advanced tools and techniques.

---

> ### Comment · Reviewer_dAYB · 2024-08-09
> **Response to Rebuttal**
>
> Thanks for your response! I'd like to see your future work to explore the sophisticated modeling nature of the whole linear models, which is closer to real data and more valuable for the interpretation of LLMs. I will keep my score.

---

### Official Review · Reviewer_xxDm · 2024-07-09

**Soundness:** 3
**Presentation:** 3
**Contribution:** 2
**Rating:** 6
**Confidence:** 3

**Summary:**

In this paper, the authors study linear transformers trained on linear regression problems and prove that each layer of every linear transformer maintains a weight vector for an underlying linear regression problem. Furthermore, the authors consider the mixed linear regression problem with varying noise levels and empirically demonstrate the flexibility of linear transformers for this problem.

**Strengths:**

**Overall well-presented**

For someone who is familiar with the field, the presentation is certainly very good. For others, it might not be super obvious that the linear transformer is trained on a bunch of other linear datasets with different noise levels. Finally, I didn't find the abbreviations before Section 2.2 particularly useful. They don't save a lot of space, but the reader has to go back to it. Not clear why the authors chose to use those.

**Relatively Interesting Insight**

I found it somewhat interesting that the transformer is doing so well in the varying noise levels scenario. But then again, might be not surprising given it had a lot of training data.

**Weaknesses:**

**Incremental Nature**

The authors build on top of a lot of existing literature. While this is not necessarily bad, the delta compared to previous work might be too small. Since I'm not an expert in this domain, I'll have to rely on the opinion of the other reviewers to confirm my understanding.

**Unclear Baselines**

I'm not sure what the baselines are for. I understand that it is useful to give an idea how well the transformer does in the task. However, the author make it look as if it was important to beat them. Furthermore, the authors claim that these methods have an advantage given that they can use matrix inversion. In my understanding, the transformer is trained on much more data which could technically used to improve the noise-level estimation for the linear models and improve their performance as well. To me it is not really clear why the authors are so defensive in this experimentation.

The authors highlight that the transformer does so much better in the varying noise levels scenario compared to the linear models. However, there was little to no effort to make the linear models good in this scenario in the first place.

**Questions:**

What are potential practical uses of this insight?

---

> ### Author Rebuttal · Authors · 2024-08-05
>
> We thank the reviewer to take their time to evaluate the paper and for valuable suggestions on improving the presentation. Indeed, the linear transformer is trained on various generated sequences of noisy linear regression. We will make it more obvious in the introduction and preliminary section. We will also move the notation from the end of Section 2.1 closer to the Theorems where they are actually being used.
>
> > I found it somewhat interesting that the transformer is doing so well in the varying noise levels scenario. But then again, might be not surprising given it had a lot of training data.
>
> When training the model, the amount of the training data is just one factor. Another big factor to consider is the capacity of the model. Not every model can fit any given function. The original motivation for this paper came from the surprising observation that Transformers have the capacity to solve certain problems provided in-context. Previous research has demonstrated that for a single noise level this ability might be due to the transformers implementing a form of gradient-based optimization on an implicitly defined objective function. For multiple noise levels previous work (Bai et al.) relied on complicated constructions with large networks and nonlinearities. In this paper we demonstrate that even _linear_ transformers with _digaonal_ attention metrics have enough capacity to solve quite complex and non-obvious problems or noisy linear regression.
>
> > ..the transformer is trained on much more data which could technically be used to improve the noise-level estimation for the linear models and improve their performance as well. To me it is not really clear why the authors are so defensive in this experimentation.
>
> Thank you for this observation. What we were trying to convey is not defensiveness, but a genuine surprise by the results that we have observed! The problem of linear regression with variable noise level, while simply stated, is practical and complex. The fact that a simple linear transformer can learn quite a sophisticated algorithm that works on par or better than many baselines that us, humans, can come up with, is quite surprising.
>
> > What are potential practical uses of this insight?
>
> We think that there are several implications of our work. One, the algorithm discovered by the linear transformer, with its adaptive rescaling and momentum-like behavior, could be directly applied to real-world noisy linear regression problems in domains such as robust control, time series analysis, or finance. Second, our work strengthens the evidence for transformers' ability to implicitly learn sophisticated algorithms, opening up exciting possibilities for automated algorithm discovery in other machine learning tasks.
>
> It would be interesting to see how variations in problem complexity and structure affect both the performance of the transformer and its ability to discover novel algorithms. This could involve exploring different data distributions, underlying function classes, and noise structures, ultimately leading to a deeper understanding of the factors influencing algorithm discovery in transformers and potentially uncovering a broader class of implicit algorithms with practical applications in various domains.

---

> > ### Comment · Reviewer_xxDm · 2024-08-10
> >
> > Thank you for the clarifications.

---

### Official Review · Reviewer_KrhZ · 2024-07-12

**Soundness:** 3
**Presentation:** 3
**Contribution:** 3
**Rating:** 7
**Confidence:** 3

**Summary:**

This paper demonstrates that linear transformers maintain a weight vector for an implicit linear regression problem. The authors provide theoretical analysis showing that (1) linear transformers maintain a linear regression model at every block and (2) a diagonal parameterization of attention heads does not compromise the expressive power of the model. The authors support their theoretical findings with experimental validation.

**Strengths:**

1. This paper extends von Oswald et al. (2023) by further investigating that each linear transformer layer maintains a weight vector that can be used for regression problems. Moreover, they show that GD++ is a second-order optimization alg for the least square problem.
2. Experiments on three different parameterization matrices are consistent with the analysis.

**Weaknesses:**

1. Limited empirical validation of Section 5.3: The results in Section 5.3 suggest that the update of y occurs every two steps. However, the paper lacks empirical studies to illustrate this phenomenon, particularly in a diagonal parameterization scenario. Can the authors provide:
a) Concrete experiments demonstrating this two-step update pattern?
b) Visualizations or quantitative analyses that show how this behavior manifests in practice?

**Questions:**

1. Comparison of attention matrices in full and diagonal parameterizations:
Given the similar loss patterns shown in Figures 2 and 3 for full and diagonal parameterizations, it raises questions about the final structure of the attention matrices. Specifically, is it possible that the full matrices converge to a near-diagonal structure during training? If so, what implications does this have for the choice between full and diagonal parameterizations in practice? If not, how do the differences in attention matrices reconcile with the similar performance observed?

---

> ### Author Rebuttal · Authors · 2024-08-05
>
> We thank the reviewer for their insightful comments and valuable feedback. We address their specific points below:
>
> > Empirical validation of Section 5.3 is lacking that the update of y occurs every two steps.
>
> Indeed, since $w_{xy}$ controls the how the current $y_t^l$ prediction affects the prediction of $x_t^{l+1}$ for the next layer, it doesn't change the prediction $y_t^{l+1}$. It takes at least two layers for this effect to reach the prediction $y_t^{l+2}$. However, it is quite challenging to empirically isolate the effect of every component, since all of the elements work simultaneously at every layer. All four terms per layer are helping to improve the loss and it is not-trivial to show the exact effect of every term empirically.
>
> > Comparison of attention matrices in full and diagonal parameterizations.
>
> We were quite surprised to see that the diagonal parametrization performs almost identical to the full one. As the reviewer has predicted, the learned attention matrices for the full case (see example in the PDF attached above) converge to the near-diagonal matrix, but with some additional structure as well. Our preliminary experiments with diagonal plus low-rank parameterizations yielded similar results to the full attention mechanism. However, given the comparable performance and interpretability of the diagonal approach, we chose to focus on this simpler diagonal model.
>
> In practice, the choice between full and diagonal parametrizations involves several factors. The diagonal approximation is easier to interpret (only 4 terms per layer!), is faster and works just as well as the full within the settings we consider. However, it is quite possible that for different distributions of $x$, $w$ or $\sigma$ beyond the ones that we consider in our paper, the difference between full and diagonal models would be more pronounced. We plan to investigate this further in future work.
>
> We will make sure to include the attention weights matrices as well as the consideration above to the final version of the paper if accepted.

---

> > ### Comment · Reviewer_KrhZ · 2024-08-08
> >
> > Thank you for the response and additional experiment. Adding more discussion or hypotheses about how the full parameterization converges with the diagonal ones would be great. Nevertheless, I like this paper and I am keeping my score.

---

### Official Review · Reviewer_WFVs · 2024-07-17

**Soundness:** 2
**Presentation:** 2
**Contribution:** 2
**Rating:** 4
**Confidence:** 2

**Summary:**

The paper tries to understand the reasons of the strong performance of Transformers. The authors study linear Transformers trained on linear regression problems. Moreover, the authors explores the problem of regression where the labels have variable noise levels.

**Strengths:**

- The problems studied are important, especially with the wide adoption of Transformers.
- The case of regression with variable noise is interesting
- Theoretical analysis is presented

**Weaknesses:**

- Although the focus is on analyzing Transformers, the paper could have benefited from adding more empirical analysis.

**Questions:**

See above

---

> ### Author Rebuttal · Authors · 2024-08-05
>
> We thank the reviewer for taking time to look at our paper. We believe that the empirical evaluation provided in the paper is thorough and is appropriate to cover the claims and contributions presented in the paper. We would love to know what kind of empirical evaluation the reviewer has in mind that we should add?

---

### Author Rebuttal · Authors · 2024-08-05

Here, we are attaching a PDF with example of learned weights for Full parametrization as requested by Reviewer KrhZ. The learned weights converge to near-diagonal matrix, which inspired us to try the diagonal parametrization.

---

### Comment · Area_Chair_fVVd · 2024-08-12
**please respond to rebuttal**

Dear reviewers,

The author-reviewer discussion period will end soon. Please make sure you read the authors' rebuttal and respond to it. If you have additional questions after reading the rebuttal please discuss with the authors. For those who have done so, thank you!

AC

---

### Decision · Program_Chairs · 2024-09-25

**Decision:**

Accept (poster)

**Comment:**

This paper provides a theoretical analysis of the performance of linear transformers under in-context learning for linear regression tasks.  The authors show that each layer of the transformer maintains an implicit linear regression model and can be interpreted as conducting a preconditioned gradient descent.  The authors also show that even using diagonal attention matrices in this case the linear transformers can still offer good expressive capability.  When dealing with mixed linear regression problems with various levels of noise, the linear transformers can yield superior performance over baselines, which can be attributed to an intrinsic optimization process with noise-aware stepsize adjustments and rescaling. The paper introduces an interesting theoretical interpretation of linear transformers under in-context learning.  The rebuttal and following discussions have cleared most of the concerns raised by the reviewers and authors also put up additional results on attention matrices.  The authors are encouraged to improve the empirical investigation to make the paper stronger.